# Glycemic control and fetal growth of women with diabetes mellitus and subsequent hypertensive disorders of pregnancy

**Mamoru Morikawa**[1]*, **Emi Kato-Hirayama**[2], **Michinori Mayama**[1], **Yoshihiro Saito**[1], **Kinuko Nakagawa**[1], **Takeshi Umazume**[1], **Kentaro Chiba**[1], **Satoshi Kawaguchi**[1], **Kazuhiko Okuyama**[2], **Hidemichi Watari**[1]

**1** Department of Obstetrics and Gynecology, Hokkaido University Graduate School of Medicine, Sapporo, Japan, **2** Department of Obstetrics and Gynecology, Sapporo City Hospital, Sapporo, Japan

* mmamoru@med.hokudai.ac.jp

**Data Availability Statement:** All relevant data are within the paper and its Supporting Information files.

## Abstract

Pregnant women with diabetes mellitus (DM) are at high risk for hypertensive disorder of pregnancy (HDP). Women with poor control DM sometimes have heavy-for-dates infants. However, women with HDP sometimes have light-for-dates infants. We aim to clarify the relationship between glycemic control and fetal growth in women with DM and/or subsequent HDP. Of 7893 women gave singleton birth at or after 22 gestational weeks, we enrolled 154 women with type 1 DM (T1DM) or type 2 DM (T2DM) whose infants did not have fetal abnormalities. Among women with T1DM or T2DM, characteristics of the three groups (with HDP, without HDP, and with chronic hypertension [CH]) were compared. No women with T1DM had CH, but 19 (17.4%) of 109 with T2DM did. HDP incidence was similar between women with T1DM (22.2%) and T2DM without CH (16.7%). Among women with T1DM, the incidences of fetal growth restriction (FGR) with and without HDP were similar. However, among women with T2DM without CH, this incidence was significantly higher among those with HDP (33.3%) than among those without HDP (5.3%), was significantly more common with HbA1c levels at first trimester $\geq$ 7.2% (33.3%) than with those < 7.2% (5.6%), and significantly more numerous without pre-pregnancy therapies for DM (23.3%) than with them (3.3%). Among women with T2DM and HDP, those with FGR had smaller placenta SDs and higher insulin dosages at delivery than those without light-for-dates. In multivariate analysis, the presence of diabetic nephropathy was a predictor of T1DM and HDP (P = 0.0105), whereas HbA1c levels $\geq$ 7.2% before pregnancy was a predictor of T2DM and HDP (P = 0.0009). Insulin dosage $\geq$ 50U/day at delivery (P = 0.0297) and the presence of HDP (P = 0.0116) independently predicted T2DM, HDP, and FGR development. Insufficient pre-pregnancy treatment of DM increased the risk of HDP.

**Funding:** The first authors received funding from the Suzuki Diabetes Foundation (Tokyo, Japan) to publish this article (for the English language reviewing charge and article processing charge).

**Competing interests:** The authors have declared that no competing interests exist.

**Abbreviations:** BMI, body mass index; CH, chronic hypertension; CI, confidence interval; DM, diabetes mellitus; FGR, fetal growth restriction; GW, gestational weeks; HbA1c, hemoglobin A1c; HDP, hypertension disorder of pregnancy; ROC, receiver operating characteristic; SD, standard deviation; T1DM, type 1 diabetes mellitus; T2DM, type 2 diabetes mellitus.

## Introduction

Pregnant women with diabetes mellitus (DM) are at high risk for hypertensive disorder of pregnancy (HDP). In the previous review of the literature, [1] the rate of preeclampsia ranged from 9% to 66% in primigravid women with type 1 DM (T1DM).

DM was found to be a risk factor for preeclampsia in a systematic review. [2] The odds ratio [95% confidence interval (CI)] of preeclampsia in pregnant women with preexisting DM versus their counterparts without DM was 3.48 (3.01–4.02) in the US [3]. Furthermore, the incidence of T2DM in Japanese populations is higher than that in the US or western European populations. In a recent study from Japan, type 2 DM (T2DM) was a risk factor for HDP, and the prevalence of HDP, gestational hypertension, and preeclampsia were 5.2–8.2, 1.8–4.4, and 0.2–9.2%, respectively. [4]

According to "hyperglycemia–hyperinsulinemia theory" (left side of Fig 1) proposed by Pedersen [5], hyperglycemia in women with DM can often induce hyperglycemia, pancreatic β cell hypertrophy, and hyperinsulinemia in the fetus as well as abnormalities such as organ immaturity and excessive weight gain. Thus, pregnant women with DM are at increased risk of having an infant that is large-for-dates infants or has macrosomia (weighing ≥4000 g). In a retrospective observational study conducted in 117,680 Japanese women without hyperglycemia who gave birth to singleton infants at 37 gestational weeks (GW) or later, a total of 1037 (0.9%) women gave birth to macrosomic neonates [6].

Pregnant women with well-controlled T1DM and normal glucose control during the first trimester and throughout pregnancy do not appear to have a higher risk of heavy-for-dates infants [7]. Many women with DM who have higher insulin resistance and whose plasma glucose levels are poorly controlled during pregnancy have infants who are heavy-for-dates. The odds ratio (and 95% CI) of macrosomic infants among women with preexisting DM in pregnancy versus their counterparts without DM was 1.91 (1.74–2.10) in the USA. [3] However, many women with DM whose plasma glucose levels were controlled too strictly had light-for-dates infants. [8] All pregnant women develop insulin resistance to some degree in response to increased plasma glucose levels. The insulin resistance is passed from mother to fetus via the placenta. The level of insulin resistance in the placenta is somewhat lower than that in the woman's bloodstream. In a previous review article, insulin resistance associated with secondary hyperinsulinemia was suspected to be the link between hypertension and DM. [9] Thus, insulin resistance and hyperinsulinemia might be the common ground for the metabolic syndrome of pregnancy-elevated blood pressure and DM. Many women with HDP have infants who are light-for-dates. According to the two-stage model of preeclampsia (right side of Fig 1), [10] stage 1 is characterized by "a poorly perfused placenta" as a result of defective remodeling of uterine spiral arteries during 8–18 GW. In pregnancy with HDP, the blood flow from mother to fetus via placenta is decreased, which results in fetal growth restriction (FGR). Approximately 15% of women diagnosed with FGR will develop preeclampsia in Japan. [11]

A brief review on preeclampsia summarized the recent work on the causes of preeclampsia, revealing a mode of maternal immune recognition of the fetus relevant to the condition [13]. In the review, preeclampsia was demonstrated to contribute to the clinical syndrome induced by a common pregnancy disorder that originates in the placenta and causes variable maternal and fetal problems.

Early-onset preeclampsia arises due to defective placentation, whereas late-onset preeclampsia may center on interactions between normal senescence of the placenta and a maternal genetic predisposition to cardiovascular and metabolic disease. The causes both placental and maternal vary among individuals [14].

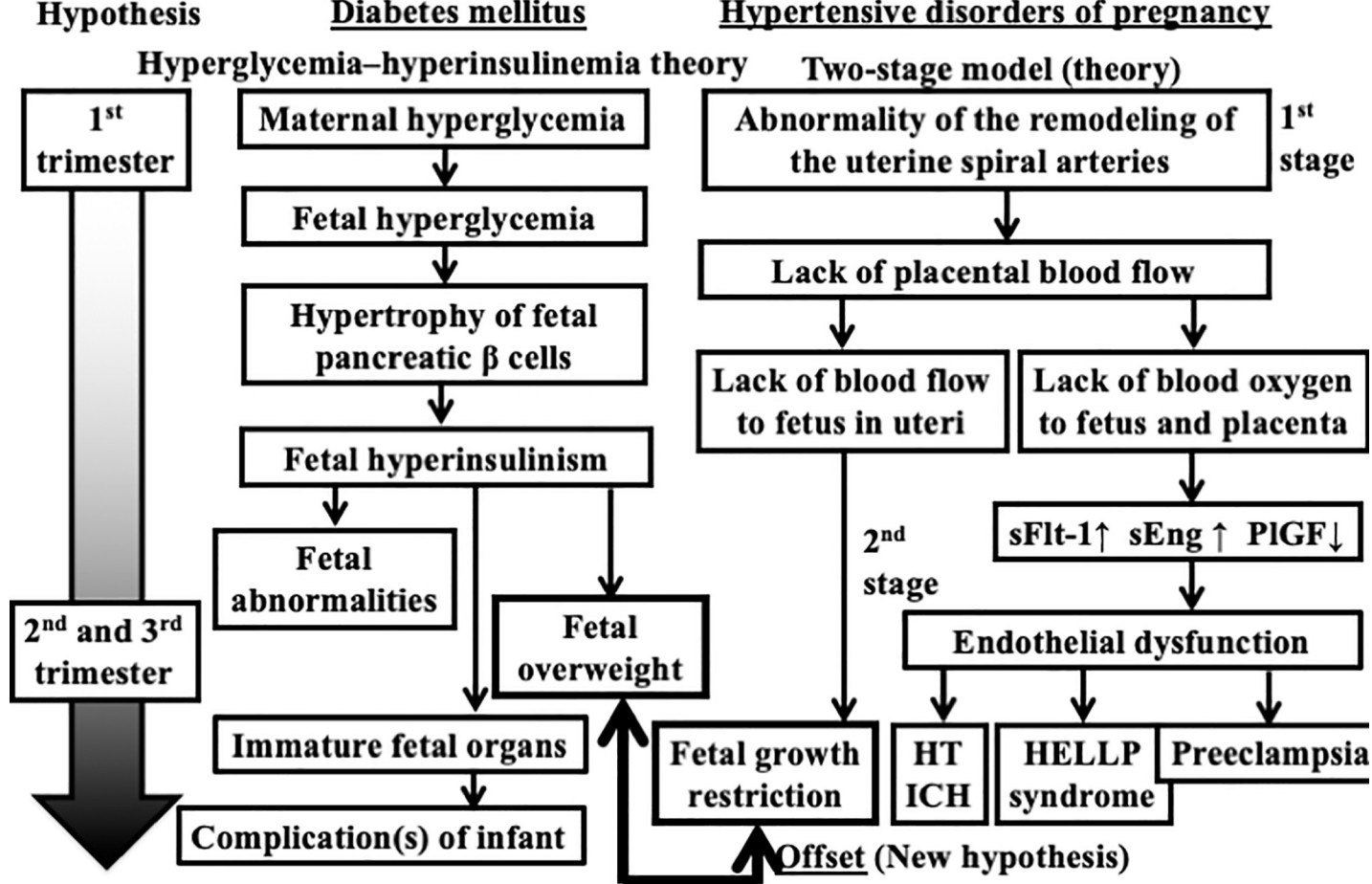

**Fig 1. Hypothesis of diabetes mellitus (DM; left side) and hypertensive disorder of pregnancy (HDP; right side).** The hyperglycemia–hyperinsulinemia theory for DM is a modified version of that described by Pedersen, [5] and the two-stage model for HDP (preeclampsia) is a modified version of that described by Roberts. [12] In this revised hypothesis, we propose that DM "normalizes" the fetal growth restriction induced by the onset of HDP. Abbreviations: GDM, gestational diabetes mellitus; HDP, hypertension disorders of pregnancy; ICH, intracerebral hemorrhage.

Women with DM would have infants who are heavier as well as a heavier placentae than women without DM [15].

In a previous report, the mean birth weight was higher in pregnant women with T1DM than in their counterparts without DM [15]. An association was identified between increased birth weight or placental weight and increased maternal BMI in pregnant women without DM, but no significant changes in birth weight or placental weight in relation to maternal BMI was noted in pregnant women with T1DM. However, women with HDP would have not only infants who are lighter but also lighter placentae [16]. In a recent report, the placental volume at 11–14 GW as determined via ultrasonography using a special technique was an independent predictor of preeclampsia and FGR related to placental insufficiency [17].

Women with DM and preeclampsia have a higher placental weight than women with pre-eclampsia alone or those without preeclampsia [16]. In addition, a recent review found that modern DM management with appropriate diet, insulin therapy, and antihypertensive treatment for chronic hypertension (CH) in pregnant women with preexisting DM can reduce the risk of pregnancy complications such as congenital malformations, fetal overgrowth, pre-eclampsia, and preterm delivery [18].

Women with both DM and HDP might be able to control their plasma glucose levels with only few insulin dosages. Did women with DM who had HDP give birth to infants who were heavy-for-dates or light-for-dates? Can glycemic control be used to predict the incidences of HDP with/without FGR in women with DM?

The aim of our study was to determine which parameters help identify women with HDP among women with DM. We examined the relation of birthweight, placenta weight, insulin dosage during the peripartum period, and hemoglobin A1c (HbA1c) levels before pregnancy or at first trimester and those at delivery in women with DM as well as in those with or without additional HDP. We thus hoped to find a way (the predictors) to detect the incidences of HDP or both HDP and FGR associated with DM.

## Materials and methods

### Study design

This retrospective cohort study was conducted in the perinatal medical centers of two institutions with maternal–fetal intensive care and neonatal intensive care units. The two institutions (Hokkaido University Hospital and Sapporo City Hospital) are located in Sapporo City with a population of 1.97 million and both institutions have specialists (medical doctors) in perinatal medicine and DM. These physicians worked in the Department of Obstetrics and Gynecology or the Department of Internal Medicine II, Hokkaido University Graduate School of Medicine.

### Inclusion and exclusion criteria

Between January 2011 and December 2018, 7893 women gave birth at or after 22 GW at the two institutions. Of these, 154 women with DM who had singleton births were enrolled. Of these, those with "overt DM during pregnancy" [19–21] were included as having T2DM. Fortunately, all women were at least 20 years old at the time of delivery. Women whose infants had fetal chromosomal abnormality were excluded as well as those who gave birth to twins or triplets.

### Diagnosis and treatment of DM during pregnancy

In the present study, the glucose metabolic disorders present before pregnancy, since early pregnancy, and both were classified into two categories: (1) T1DM and (2) T2DM, according to Japanese recommendation. [22] The women in whom DM was induced by steroid treatment were excluded.

DM was diagnosed according to the criteria [22–23] that DM was defined as (1) a fasting plasma glucose level $< 126$ mg/dL, (2) a 2-h plasma glucose level $\geq 200$ mg/dL in the 75-g oral glucose tolerance test, and (3) both random plasma glucose level $\geq 200$ mg/dL and an HbA1c level $\geq 6.5\%$.

Women with "overt DM during pregnancy" [19–21] were classified as having T2DM. The criteria for diagnosing overt DM during pregnancy were the same as those for diagnosing DM: HbA1c levels $\geq 6.5\%$ at 8 to 12 GW and before delivery.

According to Japanese recommendations for diagnosing and treating DM during pregnancy [24], insulin was the only medication used to treat DM in pregnant women in the present study. Women who used non-insulin medications before pregnancy to control plasma glucose levels switched to insulin early in their pregnancies. The targeted plasma glucose levels to treat DM were $<100$ mg/dL before meals and $<120$ mg/dL after meals.

All women in the present study who had DM were interviewed about the kind of therapy for DM that they used before and after their current pregnancies.

## Diagnosis of HDP

All women were classified into three subgroups: (1) with HDP, (2) without HDP, and (3) with chronic CH.

HDP in the present study was diagnosed as gestational hypertension and preeclampsia according to a previous Japanese criterion. [25] Hypertension was defined as a systolic blood pressure $\geq$ 140 mmHg, a diastolic blood pressure $\geq$ 90 mmHg, or both. Gestational hypertension was defined as hypertension occurring during and after 20 GW. Preeclampsia was diagnosed in women who developed both hypertension and significant proteinuria (defined as a spot-urine protein: creatinine ratio $\geq$ 0.27 or as $\geq$ 0.3 g of protein loss/24-h urine collection) $\geq$ 20 GW.

## Diagnosis of FGR or light-for-dates in infants, and evaluation basis of placenta weight and birthweight/placenta weight ratio SD

Birthweight and placenta weight were measured after deliveries. light-for-dates of infants were diagnosed in newborns on the basis of normative birthweights for Japanese newborns. [26] FGR was defined as standard deviation (SD) of birthweight $\leq$ −1.5. The placenta weight SD and birthweight/placental weight ratio SD were calculated using normative data for placenta weights and birthweights of the normal Japanese pregnant women. [27]

## Outcomes

The primary outcome was the incidence of HDP and/or FGR in pregnant women with DM. The secondary outcomes were the predictors of HDP and/or FGR in pregnant women with DM.

## Statistical analyses

Data were calculated as means ± SD or as frequencies. Statistical analyses were performed with the statistical software JMP Pro, version 14.0 (SAS Institute Inc., Cary, N.C.). Tukey–Kramer honestly significant difference tests or Student's *t* tests were used to compare the means. Fisher's exact test was used to compare categorical variables. Receiver operating characteristic (ROC) curves were used to assess the ability of parameters to differentiate incidences of HDP.

In all analyses, a p level <0.05 was used to indicate statistical significance.

## Ethics statement

This study was conducted after receiving approval from the Institutional Review Board of Hokkaido University Hospital (No. 018–297). All women provided verbal informed consent to participate in the study. The information of this clinical study is released to the public via the website of Hokkaido University Hospital according to the recommendations of the Ministry of Health, Labor, and Welfare (Japan).

## Results

### Differences in characteristics between T1DM and T2DM

Of all 154 women, 45 women (29.2%) had T1DM and 109 women (70.8%) had T2DM. Thus, incidence rates of glucose metabolic disorders before pregnancy were 2.0% (154/7893). No

**Table 1. Characteristics of pregnant women with and without hypertension with T1DM and T2DM during pregnancy.**

| Characteristic | T1DM [c] | | | T2DM | | | |
|---|---|---|---|---|---|---|---|
| | Overall (*n* = 45, 100%) | T1DM+HDP (*n* = 10, 22.2%) | T1DM alone (*n* = 35, 77.8%) | Overall (*n* = 109, 100%) | T2DM+ HDP (*n* = 15, 13.8%) | T2DM alone (*n* = 75, 68.8%) | T2DM+ CH (*n* = 19, 17.4%) |
| Age (years) | 30.9 ± 0.8 [a] | 30.3 ± 4.8 | 31.1 ± 5.2 | 33.5 ± 0.5 | 32.3 ± 5.7 | 33.3 ± 5.8 | 34.9 ± 4.5 |
| Primipara (%) | 27 (60.0%) | 7 (70.0%) | 20 (57.1%) | 49 (45.0%) | 8 (53.3%) | 32 (42.7%) | 9 (47.4%) |
| Preeclampsia (%) | 6 (13.3%) | 6 (60.0%) | — | 10 (9.2%) | 7 (46.7%) | — | 3 (15.8%) |
| Gestational age at delivery (weeks) | 37.7 ± 0.4 | 37.4 ± 2.4 | 37.8 ± 1.8 | 37.2 ± 0.3 | 36.3 ± 2.9 | 37.6 ± 2.7 | 36.1 ± 4.1 |
| Preeclampsia (%) | 6 (13.3%) | 6 (60.0%) | — | 10 (9.2%) | 7 (46.7%) | — | 3 (15.8%) |
| Diabetic nephropathy (%) | 6 (13.3%) | 4 (40.0%) [b] | 2 (5.7%) | 12 (11.0%) | 4 (26.7%) | 6 (8.0%) | 2 (10.5%) |
| Diabetic retinopathy (%) | 8 (17.8%) | 1 (10.0%) | 7 (20.0%) | 13 (11.9%) | 3 (20.0%) | 6 (8.0%) | 4 (21.1%) |
| BMI before pregnancy | 22.5 ± 0.8 [a] | 22.6 ± 4.2 [a] | 22.4 ± 3.1 [a] | 28.9 ± 0.5 | 30.7 ± 4.9 | 27.8 ± 5.8 | 31.8 ± 4.8b |
| BMI at delivery | 26.6 ± 0.7 [a] | 28.0 ± 3.5 [a] | 26.2 ± 2.9 [a] | 32.0 ± 0.5 | 34.6 ± 5.2 [b] | 30.8 ± 5.4 | 34.7 ± 5.3 [b] |
| ΔBMI during pregnancy | 4.15 ± 0.32 [a] | 5.45 ± 2.01 [b] | 3.77 ± 1.98 | 3.12 ± 0.20 | 3.85 ± 2.11 | 3.03 ± 2.18 | 2.94 ± 1.91 |
| HbA1c during first trimester (%) | 7.46 ± 1.64 | 8.54 ± 2.24 [b] | 7.15 ± 1.31 | 7.30 ± 1.56 | 7.91 ± 1.22 | 7.11 ± 1.62 | 7.57 ± 1.42 |
| HbA1c at delivery (%) | 6.55 ± 0.90 [a] | 7.07 ± 1.17b | 6.41 ± 0.76 | 6.19 ± 0.88 | 6.65 ± 1.48 | 6.12 ± 0.70 | 6.12 ± 0.87 |
| ΔHbA1c during pregnancy (%) | - 0.91 ± 1.24 | - 1.47 ± 1.95 | - 0.75 ± 0.92 | - 1.11 ± 1.50 | - 1.26 ± 1.33 | - 1.00 ± 1.54 | - 1.45 ± 1.45 |
| Treatment of DM before pregnancy (%) | 41 (91.1%) [a] | 8 (80.0%) | 33 (94.3%) | 37 (33.9%) | 1 (6.7%) [b] | 29 (38.7%) | 7 (36.8%) |
| ID during first trimester (U/day) | 31.6 ± 21.4 [a] | 26.4 ± 24.7 [a] | 33.1 ± 20.5 [a] | 8.1 ± 16.1 | 0.8 ± 3.1 | 9.6 ± 18.0 | 7.9 ± 12.5 |
| ID at delivery (U/day) | 51.4 ± 20.2 | 41.1 ± 13.2 [a] | 54.4 ± 21.1 | 63.1 ± 48.0 | 93.4 ± 63.1 [b] | 56.5 ± 42.8 | 64.9 ± 47.4 |
| Δ ID during pregnancy (U/day) | 19.9 ± 22.2a | 14.7 ± 25.7 [a] | 21.3 ± 21.3 [a] | 55.0 ± 47.8 | 92.6 ± 63.2 [b] | 46.9 ± 40.9 | 47.3 ± 10.9 |
| Cesarean section (%) | 21 (46.7%) | 6 (60.0%) | 15 (42.9%) | 65 (59.6%) | 12 (80.0%) [b] | 38 (50.7%) | 15 (78.9%) [b] |
| FGR (%) | 1 (2.2%) | 0 (0.0%) | 1 (2.9%) | 12 (11.0%) | 5 (33.3%) [b] | 4 (5.3%) | 3 (15.8%) |
| Birthweight (g) | 3145 ± 581a | 2828 ± 646 [b] | 3236 ± 537 [a] | 2828 ± 769 | 2629 ± 1012b | 2943 ± 640 | 2589 ± 959 |
| Birthweight SD | 1.11 ± 1.27 [a] | 0.47 ± 1.51 | 1.33 ± 1.15 [a] | 0.48 ± 1.56 | 0.26 ± 2.41 | 0.59 ± 1.34 | 0.22 ± 1.62 |
| Placenta weight (g) | 621 ± 121 | 561 ± 135 | 639 ± 113 | 586 ± 164 | 546 ± 205 | 602 ± 150 | 555 ± 182 |
| Placenta weight SD | 0.65 ± 0.95 | 0.13 ± 1.12 | 0.80 ± 0.83 | 0.39 ± 1.27 | 0.07 ± 1.67 | 0.49 ± 1.16 | 0.26 ± 1.34 |
| BW/PW ratio | 5.10 ± 0.77 | 5.09 ± 0.77 | 5.10 ± 0.78 | 4.89 ± 0.90 | 4.81 ± 0.85 | 4.97 ± 0.85 | 4.62 ± 1.13 |
| BW/PW ratio SD | 0.19 ± 0.84 | 0.32 ± 0.72 | 0.15 ± 0.88 | 0.05 ± 0.97 | 0.17 ± 0.80 | 0.03 ± 1.01 | 0.02 ± 0.93 |
| Stillbirth or END (%) | 1 (2.2%) | 0 (0.0%) | 1 (2.9%) | 2 (1.8%) | 0 (0.0%) | 2 (2.7%) | 0 (0.0%) |

Data are expressed as means ± standard deviations (SDs). Hypertensive disorders of pregnancy (HDP) were defined as the new onset of hypertension during pregnancy. BMI, body mass index; ΔBMI, change in BMI; BW/PW, birthweight/placenta weight; END, early neonatal death; HbA1c, hemoglobin A1c; ΔHbA1c, change in hemoglobin A1c; HT, hypertension; ID, insulin dose; ΔID, change in insulin dose; T1DM, diabetes mellitus type 1; T2DM, diabetes mellitus type 2.

[a] p < 0.05 (T1DM versus T2DM).

[b] p < 0.05 (HDP versus non-HDP in the same group).

[c] No woman with T1DM had chronic hypertension.

women with T1DM had CH, but 19 women (17.4%) with T2DM did. The incidence of HDP among women with T1DM (n = 10, 22.2%) was similar to that among women with T2DM (n = 15, 13.8%). Furthermore, the incidence of preeclampsia among women with T1DM (13.3%) was similar to that among women with T2DM (9.2%).

The characteristics of women with T1DM and T2DM are shown in Table 1. The frequency of women with diabetic nephropathy or diabetic retinopathy among those with T1DM was

similar to those with T2DM. The body mass index (BMI) before pregnancy, BMI at delivery, and amount of increase of BMI during pregnancy was significantly higher among women with T1DM than among those with T2DM ($p < 0.0001$, $p < 0.0001$, and $p = 0.0072$, respectively). However, HbA1c at first trimester, HbA1c at delivery, and the amount of increase of HbA1c during pregnancy were similar between women with T1DM and those with T2DM. The treatment of DM before pregnancy among women with T2DM was less common than among those with T1DM ($p < 0.0001$). The insulin dosage during the first trimester was significantly higher among women with T1DM than among those with T2DM ($p < 0.0001$); however, the insulin dosage at delivery was similar between women with T1DM and those with T2DM. Birthweight and SD of birthweight were significantly higher among women with T1DM than among those among those with T2DM ($p = 0.0174$ and $p = 0.0133$, respectively); however, the frequency of FGR was similar between women with T1DM and those with T2DM. Moreover, placenta weight and placenta weight SDs at delivery were similar between women with T1DM and those with T2DM.

## Differences in glycemic control, fetal growth, and placenta weight among pregnant women with T1DM+HDP and those with T1DM alone (without HDP)

Table 1 lists the characteristics of the two groups of women. The frequency of women with diabetic nephropathy in the T1DM+HDP group was significantly higher than those in the T1DM group ($p = 0.0164$); however, the frequency of women with diabetic retinopathy was similar between the two groups. BMIs before pregnancy and at delivery were similar between the T1DM+HDP and T1DM alone groups; however, the increase in BMI during pregnancy in the T1DM+HDP group was significantly higher than that in the T1DM alone group ($p = 0.0233$).

In contrast, HbA1c levels during the first trimester and at delivery in the T1DM+HDP group were significantly higher than those in the T1DM alone group ($p = 0.0166$); however, the increase in HbA1c levels during pregnancy in the T1DM+HDP group was similar to that in the T1DM alone group. Insulin dosages during first trimester and at delivery, and the increase in insulin dosage increases during pregnancy were similar between the two groups. Birthweight in the T1DM+HDP group was lower than that in the T1DM alone group ($p = 0.0491$): however, birthweight SDs, placenta weights, and placenta weight SDs were similar between the two groups. The frequency of FGR in women with T1DM+HDP was similar to that in women with T1DM alone.

## Differences in glycemic control, fetal growth, and placental weight among pregnant women with T2DM+HDP, those with T2DM alone, and those with T2DM+CH

Table 1 shows the characteristics of the three groups in women with T1DM and T2DM.

The frequency of diabetic nephropathy or diabetic retinopathy was similar among the three groups.

BMIs before pregnancy and the increase in BMI during pregnancy were similar between the women with T2DM+HDP and those with T2DM alone; however, BMIs at delivery were significantly higher in women with T2DM+HDP than in those with T2DM alone ($p = 0.0274$). Furthermore, BMIs before pregnancy and at delivery were significantly higher in women with T2DM+CH than in those with T2DM ($p = 0.0073$ and $p = 0.0060$, respectively) but were similar to those with T2DM+HDP.

HbA1c levels during the first trimester and at delivery and the increase in BMI during pregnancy were similar among the three groups. The treatment of DM before pregnancy in the

T2DM+HDP group was less common than that in the T2DM alone group (p = 0.0164) or in the T2DM+CH group (p = 0.0294). Insulin dosages at delivery in the T2DM+HDP group were significantly higher than those in the T2DM alone group (p = 0.0063). Thus, the amount of increase of insulin dosage during pregnancy was significantly higher in the T2DM+HDP group than in the T2DM alone group (p = 0.0027).

The birthweight in T2DM+HDP group was lower than that in the T2DM alone group (p = 0.0022), and the frequency of FGR in the T2DM+HDP group was higher than that in the T2DM group (p = 0.0057), however, birthweight SD, placenta weight, and placenta weight SD were similar between the two groups. Birthweight, birthweight SD, placenta weight, and placenta weight SD in the T2DM+CH group were similar to those in the T2DM+HDP group and to those in the T2DM alone group.

## Relationship between incidences of HDP and pre-pregnancy glycemic control / glycemic control at delivery

Of the 90 women with T2DM but without CH, 15 (16.7%) had HDP (see Table 1). We considered the relation between incidences of HDP and pre-pregnancy glycemic control among 135 women: the 45 with T1DM (i.e., T1DM+HDP or T1DM alone groups) and the 90 with T2DM without CH (i.e., T2DM+HDP or T2DM alone groups).

According to ROC curves, there was a relationship between HbA1c level before pregnancy or at first trimester and incidences of HDP for all 135 women (cutoff value of HbA1c level before pregnancy, $\geq$ 7.2%; AUC = 0.704, p = 0.0239), for all women with T1DM (that, $\geq$ 6.8%; p = 0.0239), and among all women with T2DM without CH (that, $\geq$ 7.2%; AUC = 0.706, p = 0.0864), respectively. Moreover, there were no relations between insulin dosage during the first trimester and incidences of HDP among all 135 women or among the 45 women with T1DM. However, among the 90 women with T2DM without CH, there was a relation between insulin dosage during the first trimester and incidences of HDP (cutoff value of insulin dosage before pregnancy was 0 IU/day; AUC = 0.664, p = 0.0064).

The relations between incidence of HDP and HbA1c values during first trimester, therapy for DM before pregnancy and at delivery are shown in Fig 2.

Among women with T1DM, HDP occurred significantly more among women with HbA1c values before pregnancy or at first trimester $\geq$ 6.8% (33.3%) than among those with HbA1c values < 6.8% (5.6%; p = 0.0343, Fig 2A). However, among women with HbA1c levels $\geq$ 6.8%, the odds ratio for HDP was 8.50 (95% CI, 0.97–74.4, Fig 2A). Among women with T2DM without CH, the frequency of HDP was significantly higher among women with HbA1c values before pregnancy or at first trimester $\geq$ 7.2% (33.3%) than that among those with HbA1c values < 7.2% (5.6%; p = 0.0009, Fig 2B). Among women with HbA1c values $\geq$ 7.2%, the odds ratio was 8.50 (95% CI, 2.19–33.0, Fig 2B). Among women with T1DM, the frequency of HDP among those who did not receive therapy for DM before pregnancy was similar to that among those who did (Fig 2C). However, among women with T1DM and T2DM without CH, the frequency of HDP was higher among those who did not receive therapy for DM before pregnancy than that among those who did. In particular, among women with T2DM without CH, the frequency of HDP was significantly higher among women who did not receive therapy for T2DM without CH before pregnancy (23.3%) than that among those who did (3.3%; p = 0.0168, Fig 2D). The odds ratio for HDP among women who received therapy for T2DM without CH before pregnancy was 8.83 (95% CI, 1.10 to 70.7, Fig 2D).

We considered the relation between the incidences of HDP and glycemic control at delivery among 135 women (45 with T1DM and 90 with T2DM without CH). According to the ROC curves, no relation was observed between insulin dosages at delivery and HDP incidences for

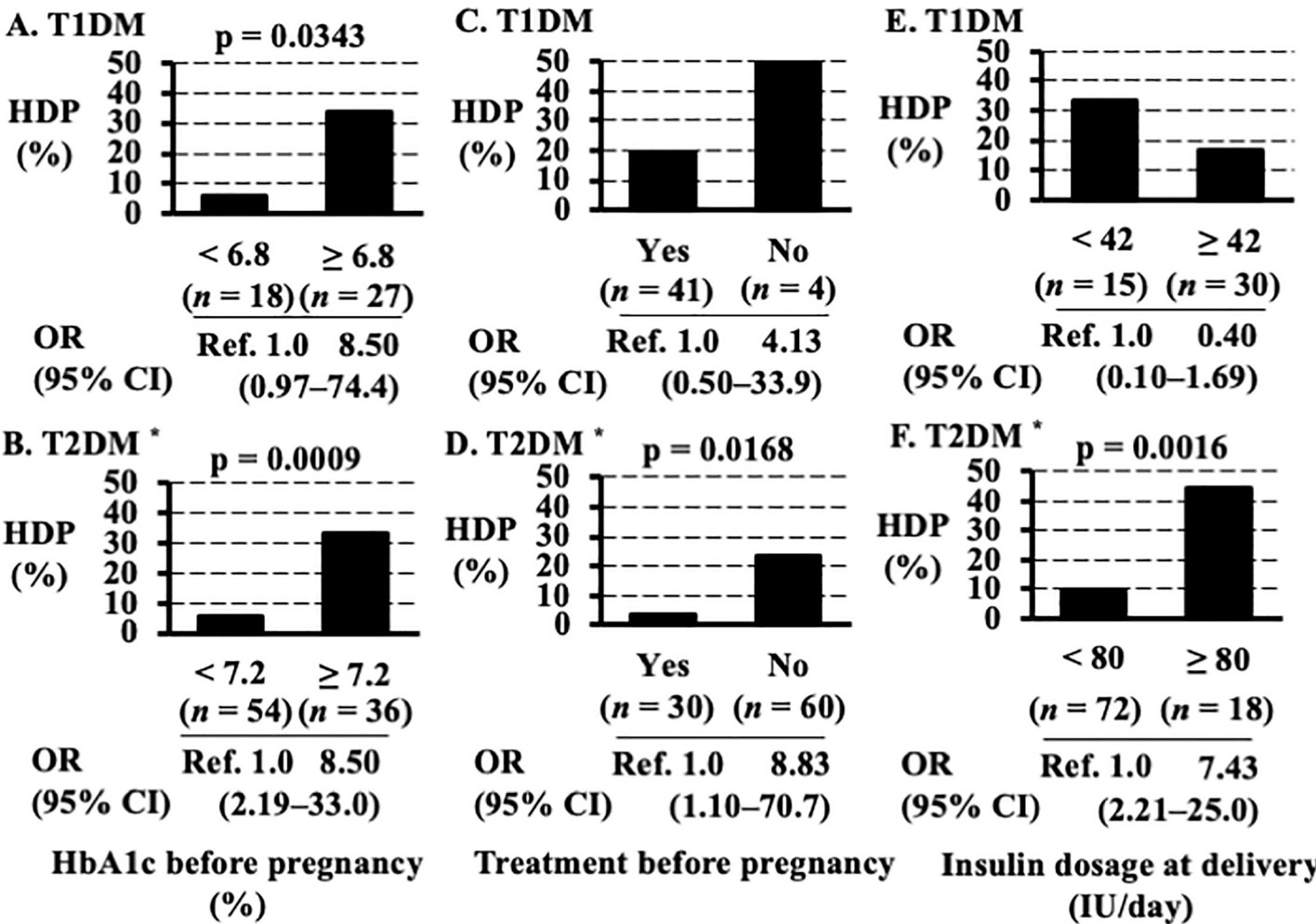

**Fig 2. Differences in the incidence of HDP among women with HbA1c values above or below cutoffs during first trimester and between women who did or did not receive therapy for diabetes mellitus before pregnancy and at delivery.** (A) Among women with T1DM, the differences in incidence of HDP among women with HbA1c values above or below the cutoff during the first trimester. (B) Among women with T2DM without CH, the differences in incidence of HDP among women with HbA1c values above or below the cutoff during the first trimester. (C) Among women with T1DM, the differences in incidence of HDP among women who did or did not receive therapy for diabetes mellitus before pregnancy. (D) Among women with T2DM without CH, the differences in incidence of HDP among women who did or did not receive therapy for diabetes mellitus before pregnancy. (E) Among women with T1DM, the differences in incidence of HDP among women with insulin dosage at delivery for diabetes mellitus above or below the cutoff. (F) Among women with T2DM without CH, the differences in the incidence of HDP among women with insulin dosage at delivery for diabetes mellitus above or below the cutoff. Abbreviations: HDP; hypertension disorder of pregnancy, HbA1c; hemoglobin A1c, T1DM, type 1 diabetes mellitus; T2DM without CH, type 2 diabetes mellitus but without chronic hypertension; CI, confidence interval; OR, odds ratio; Ref, reference.

any of the 135 women (cutoff value of insulin dosage at delivery, $\geq$ 80 IU/L; p = 0.0866). However, we found a relation between insulin dosages at delivery and HDP incidences for all women with T1DM (dosage $\geq$ 42 IU/L; AUC = 0.671, p = 0.0361), and among all women with T2DM without CH (dosage, $\geq$80 IU/L; AUC = 0.690, p = 0.0124), respectively. Among women with T1DM, HDP frequencies were similar between women who received therapy with insulin dosages $\geq$ 40 IU/L for T1DM at delivery (16.7%) and those who received therapy with insulin dosages < 40 IU/L for T1DM at delivery (9.7%, Fig 2E). In particular, among women with T2DM without CH, HDP frequency was significantly higher among women who received therapy with insulin dosages $\geq$ 80 IU/L for T2DM without CH at delivery (44.4%) than among those who received therapy with insulin dosages < 80 IU/L for T2DM without CH at delivery (9.7%; p = 0.0016, Fig 2F). The odds ratio for HDP among women who received

therapy with insulin dosages ≥ 80 IU/L for T2DM without CH at delivery was 7.43 (95% CI, 2.21–25.0, Fig 2F).

## Relationships between incidences of HDP according to glycemic control and birthweight or placental weight

Relationships between fetal weight, placental weight, and insulin dosage before pregnancy and at delivery among women with HDP or CH were shown in Fig 3.

The insulin dosage before pregnancy by birthweight ratio in the T1DM+HDP group was higher than in the T2DM group (p = 0.0009 against those in the T2DM+HDP group; p = 0.0031 against those in the T2DM alone group, and p = 0.0194 against those in the T2DM +CH group; see Fig 3A). The same ratio in the T1DM group was higher than that in the T2DM group (p < 0.0001 against that in the T2DM+HDP group, p < 0.0001 against that in the T2DM alone group, and p = 0.0002 against that in the T2DM+CH group; see Fig 3A). Moreover, the insulin dosage before pregnancy by placental weight ratio in the T1DM+HDP group was higher than that in the T2DM group (p = 0.0002 against that in the T2DM+HDP group, p = 0.0007 against that in the T2DM alone group, and p = 0.0059 against that in the T2DM+CH group; see Fig 3B). In addition, the ratio in the T1DM alone group was higher than that in T2DM group (p < 0.0001 against that in the T2DM+HDP group, p < 0.0001 against that in the T2DM alone group, and p < 0.0001 against that in the T2DM+CH group; see Fig 3B).

The insulin dosage at delivery by birthweight ratio in the T2DM+CH group was higher than that in the T2DM alone group (p = 0.0006), that in the T1DM+HDP group (p = 0.0014), and that in the T1DM alone group (p = 0.0002; Fig 3C). The insulin dosage at delivery by birthweight ratio in the T2DM+HDP group was higher than that in the T2DM+CH group (p = 0.0181), that in the T1DM+HDP group (p = 0.0329), and that in the T1DM alone group (p = 0.0045) (Fig 3C). Moreover, the insulin dosage at delivery by placental weight ratio in the T2DM+HDP group was higher than that in the T2DM alone group (p = 0.0011), that in the T1DM + HDP group (p = 0.0033), and that in the T1DM alone group (p = 0.0003; Fig 3D).

## Relationship between incidences of HDP or FGR according to glycemic control and placental weight

The frequency of FGR in the T1DM+HDP group (0.0%) was similar to that in T1DM group (2.9%); however, those of women with both T2DM+HDP group (33.3%) were higher than those of women with T2DM alone group (5.3%, p = 0.0057, respectively). Among women with HDP, the odds ratio for FGR was 8.88 (95% CI, 2.04–38.7).

We considered the relationship between incidences of FGR and placenta weight, glycemic control among women with T2DM in Fig 4.

Placenta weights SD of mothers of infants with FGR were significantly lower than those of mothers of infants without FGR in the T2DM+HDP group (−1.70 ± 0.70 *versus* 0.96 ± 1.24, p = 0.0007), and in the T2DM+CH group (−1.45 ± 0.96 versus 0.58 ± 1.16, p = 0.0114) (Fig 4A). HbA1c levels during the first trimester and at delivery were similar between mothers of infants with FGR and mothers of infants without FGR and between women with HDP and those without HDP (Fig 4B). In the T2DM alone group, the insulin dosages during the first trimester were higher for mothers of infants without FGR (10.1 ± 18.4 U/day) than those for mothers of infants with FGR (0.0 ± 0.0 U/day; p < 0.0001), but first-trimester insulin dosages for mothers of infants without FGR (1.2 ± 3.8 U/day) were similar to those for mothers of infants with FGR (0.0 ± 0.0 U/day) (Fig 4C). The insulin dosages at delivery for women with HDP only (121 ± 58.3 U/day) were higher than those for women with both HDP and FGR

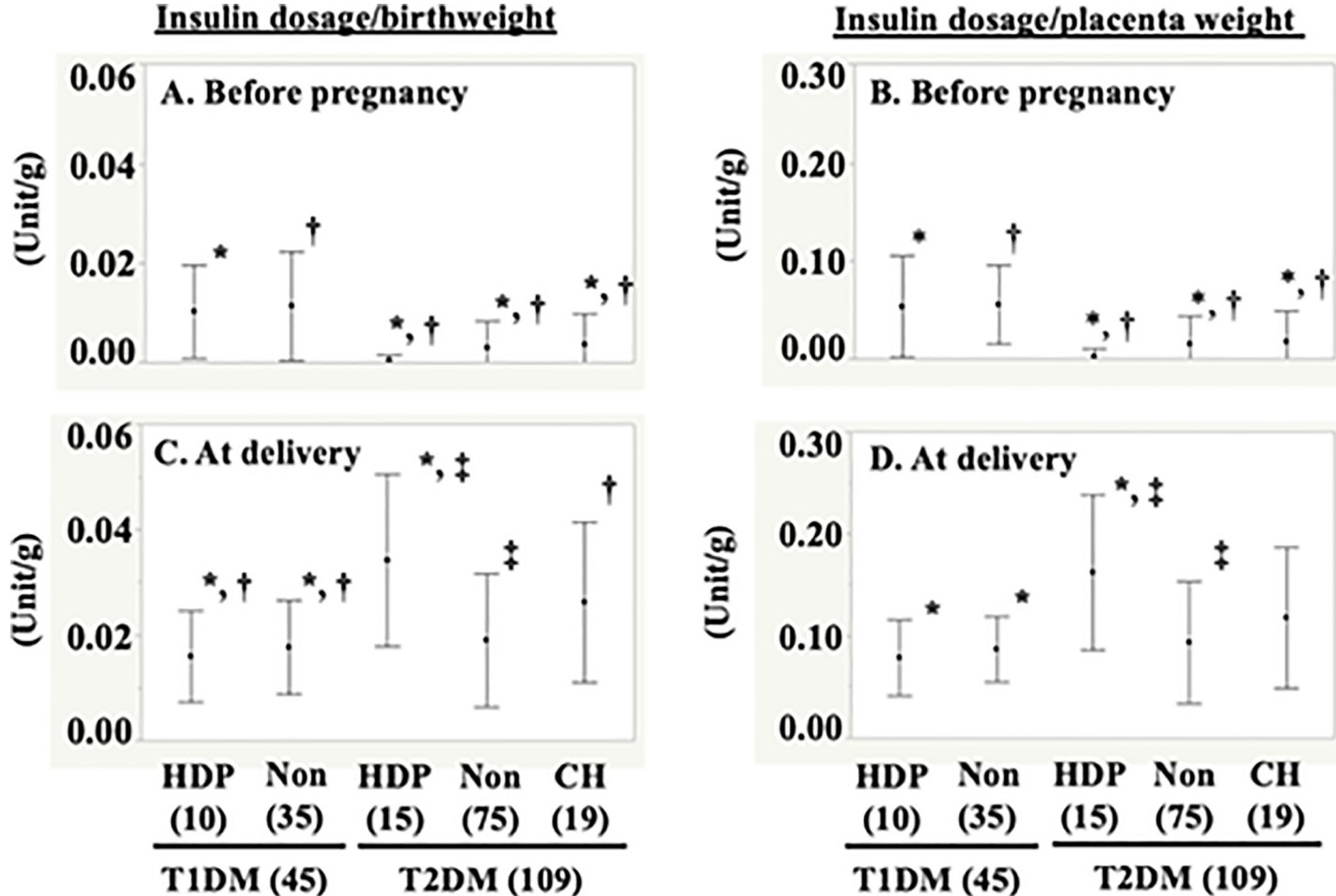

**Fig 3. Relationships between fetal weight, placental weight, and insulin dosage before pregnancy and at delivery among women with HDP or CH.** (A) Insulin dosage before pregnancy/fetal weight ratios. (B) Insulin dosage before pregnancy/placental weight ratios. (C) Insulin dosage at delivery/fetal weight ratios. (D) Insulin dosage at delivery/placental weight ratios. Abbreviations: HDP; hypertension disorder of pregnancy, Non; without hypertension disorder of pregnancy, CH; chronic hypertension. [a] p < 0.05 versus women with T1DM, [b] p < 0.05 versus women with T1DM, and [c] p < 0.05 versus women without HDP among women with T2DM.

(37.8 ± 21.7 U/day; p = 0.0016), and those for women without both diseases (57.9 ± 43.4 U/day) were higher than those for mothers of infants with FGR but without HDP (32.0 ± 14.8 U/day; p = 0.0257) (Fig 4D). However, among women who did not receive therapy for T2DM before pregnancy, the frequency of FGR was 0.0%, whereas in the T2DM+HDP group who did receive therapy, the frequency of FGR was 35.7%. In the T2DM alone group who received therapy, the frequency of FGR was 8.7%. Moreover, in the T2DM+CH group, insulin dosages at delivery for women without FGR were similar to those with FGR (71.6 ± 48.9 U/day versus 29.3 ± 5.5 U/day).

## Relationship between glycemic control and FGR in women with DM and/or HDP according to multivariate analysis results

Table 2 shows the predictors of HDP, FGR, and HDP+FGR identified using multivariate analysis with the following variables: primipara or multipara (≥ 35 years or < 35 years), BMI before pregnancy (≥ 25 or < 25), presence or absence of diabetic nephropathy, treatment or no treatment before pregnancy, HbA1c levels before pregnancy or at first trimester (≥ 6.8%

## T2DM

### A. Placenta weight SD

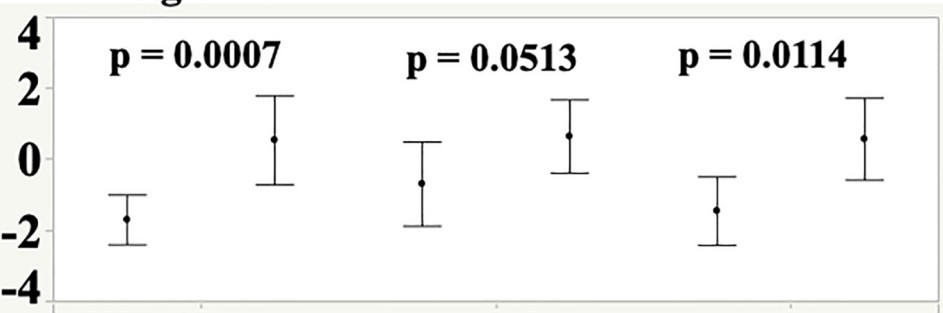

### B. HbA1c at first trimester

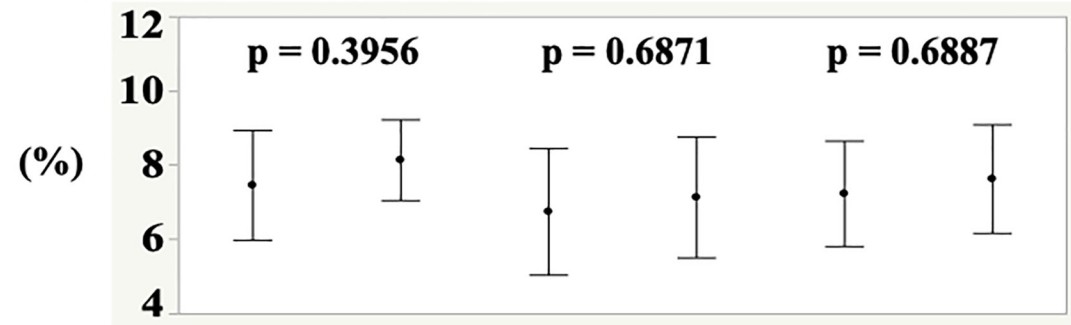

### C. Insulin dosage before pregnancy

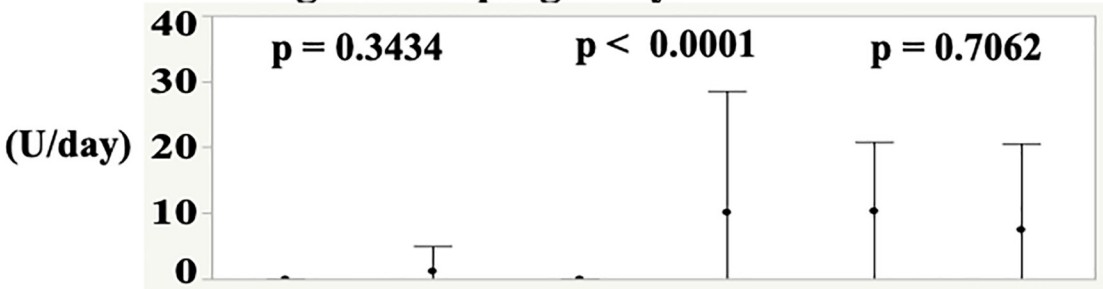

### D. Insulin dosage at delivery

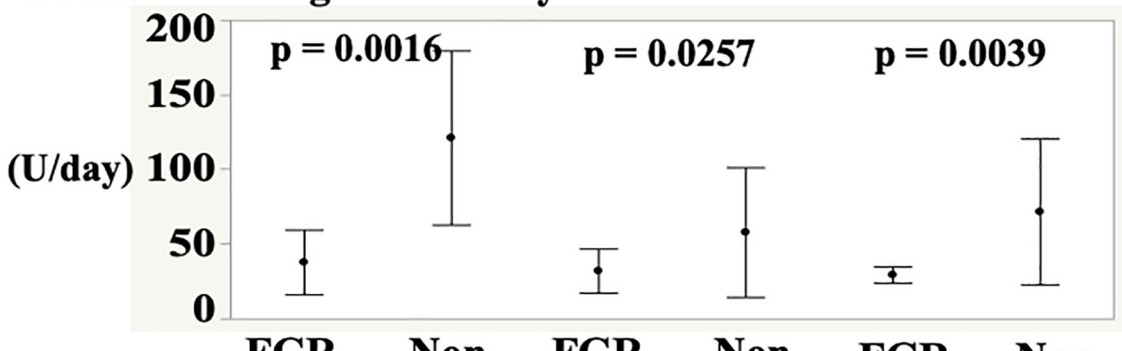

**Fig 4. Relationship between incidences of HDP and FGR according to glycemic control and placenta weight among women with T2DM.** (A) Placenta weight. (B) HbA1c at first trimester. (C) Insulin dosage before pregnancy. (D) Insulin dosage at delivery. Abbreviations: HDP; hypertension disorder of pregnancy, FGR; fetal growth restriction, T2DM; type 2 diabetes mellitus, HbA1c; hemoglobin A1c, Non; without fetal growth restriction, Non-HDP; without hypertension disorder of pregnancy, CH; chronic hypertension.

or < 6.8% in women with T1DM, ≥ 7.2% or < 7.2% in women with T2DM without CH), presence or absence of HDP, insulin dosage at delivery (≥ 42 IU/L or < 42 IU/L for predicting HDP in women with T1DM, ≥ 80 IU/L or <80 IU/L for predicting HDP in women with T2DM without CH, ≥ 50 IU/L or < 50 IU/L for predicting FGR in women with T2DM without CH, and ≥ 33 IU/L or < 33 IU/L for predicting FGR in women with T2DM+CH).

The predictors of HDP incidence included the presence of diabetic nephropathy in women with T1DM (p = 0.0105) and HbA1c levels before pregnancy or at first trimester ≥ 7.2% in women with T2DM without CH (p = 0.0009). Furthermore, the independent predictors of FGR in women with T2DM without CH were insulin dosage at delivery ≥ 50 U/day (p = 0.0297) and the HDP incidence (p = 0.0116). The predictor of FGR among women with T2DM+CH was insulin dosage at delivery ≥ 33 U/day only according to univariate analysis (p = 0.0274).

We found no predictors of the incidence of HDP+FGR in women with T2DM+CH by multivariate analysis. However, according to ROC curves, a correlation between placental weight SD and the incidence of HDP+FGR was noted for all 90 women with T2DM without CH

**Table 2. The predictors of the incidences of HDP and/or FGR among women with T1DM or T2DM without CH.**

| | Odds ratio | 95% CI | Sensitivity | Specificity | PPV | NPV | Univariate analysis P value | Multivariate analysis P value |
|---|---|---|---|---|---|---|---|---|
| *Incidences of HDP among women with T1DM* | | | | | | | | |
| Present of diabetic nephropathy | 11.0 | 1.63–74.1 | 0.667 | 0.846 | 0.400 | 0.943 | 0.0164 | 0.0105 |
| HbA1c levels at first trimester ≥ 6.4% | 8.50 | 0.97–74.4 | 0.900 | 0.496 | 0.333 | 0.944 | 0.0343 | |
| *Incidences of HDP among women with T2DM without CH* | | | | | | | | |
| HbA1c levels at first trimester ≥ 7.2% | 8.50 | 2.19–33.0 | 0.800 | 0.680 | 0.333 | 0.944 | 0.0009 | 0.0009 |
| Without treatment before pregnancy | 8.83 | 1.10–70.7 | 0.933 | 0.387 | 0.233 | 0.967 | 0.0168 | |
| Insulin dosage at delivery ≥ 80 U/day | 7.43 | 2.21–25.0 | 0.533 | 0.867 | 0.444 | 0.903 | 0.0016 | |
| *Incidences of FGR among women with T2DM without CH* | | | | | | | | |
| Without treatment before pregnancy | 340 | 19.0–6077 | 1.00 | 0.769 | 0.850 | 1.00 | 0.0266 | |
| Insulin dosage at delivery ≥ 50 U/day | 12.3 | 1.46–103 | 0.889 | 0.605 | 0.200 | 0.980 | 0.0274 | 0.0297 |
| Incidences of HDP | 8.88 | 2.04–38.7 | 0.556 | 0.877 | 0.333 | 0.947 | 0.0057 | 0.0116 |
| *Incidences of FGR among women with T2DM +CH* | | | | | | | | |
| Insulin dosage at delivery ≥ 33 U/day | 42.0 | 1.49–1186 | 1.00 | 0.875 | 0.600 | 1.00 | 0.0274 | |

T1DM, type 1 diabetes mellitus; T2DM, type 2 diabetes mellitus; HDP, hypertension disorder of pregnancy; CH, chronic hypertension; FGR, fetal growth restriction; HbA1c, hemoglobin A1c; SD, standard deviation; CI, confidence interval; PPV, positive predictive value; NPV, negative predictive value

(cutoff value of placental weight SD, −6.9; AUC = 0.960; p < 0.0001). Among the women with T2DM without CH, HDP+FGR occurred significantly more often in those with placental weight SDs < −6.9 (5/17, 29.4%) than in those with placental weight SDs ≥ −0.69 (0/73, 0.0%; p = 0.0001). Therefore, among the women with placental weight SDs < −6.9, the odds ratio for the incidence of HDP+FGR was 60.8 (95% CI, 3.12–1186). Furthermore, according to the ROC curves, we found a relationship between placental weight SD and the incidence of FGR (with or without HDP) for all 90 women with T2DM without CH (cutoff value of placental weight SD, −6.9; AUC = 0.881; P < 0.0001). Among women with placental weight SDs < −6.9, the odds ratio for HDP + FGR was 64.0 (95% CI, 7.15–673).

## Discussion

The findings of this study emphasized the following five points. First, no women with T1DM had CH; however, 17.4% of women with T2DM did have CH. The frequency of HDP among women with T1DM (22.2%) was similar to that among those with T2DM but without CH (16.7%). Second, the frequency of FGR among women with T1DM and HDP was similar to that among those with T1DM but without HDP, however the frequency of women with HDP was approximately nine-fold higher among women with T2DM and HDP than that among women with T2DM but without HDP. Third, among women with T2DM but without CH, the incidence of HDP was approximately eightfold higher among women with HbA1c levels ≥ 7.2% that among those with HbA1c levels < 7.2%. Fourth, among women with both T2DM and HDP or CH, mothers of infants with FGR had a significantly lighter placenta and a significantly higher insulin dosage at delivery. Fifth, using multivariate analysis, the predictors of incidence of HDP were the presence of diabetic nephropathy among women with T1DM and HbA1c before pregnancy ≥ 7.2% among those with T2DM without CH; furthermore, the predictors of incidences of FGR among women with T2DM without CH were insulin dosage at delivery ≥ 50 U/day and the incidences of HDP, independently.

To our knowledge, this is the first study to demonstrate the relationship between DM and incidences of HDP in relation to birthweight SD, placenta weight and HbA1c level during the first trimester and in relation to insulin dosage before pregnancy.

We hypothesize that the FGR induced by HDP might "offset" fetal overweight (macrosomia or heavy-for-dates infants) induced by DM (Fig 1).

In another study, older maternal age, obesity, and/or vascular diseases were determined as risk factors of preeclampsia. [28] In the other study, not all pregnant women with obesity developed preeclampsia; however, those with the most metabolic anomalies had the highest incidence of preeclampsia. [29] Furthermore, the study reported that metabolic anomalies, such as increased circulating leptin, glucose, insulin, and lipids, were said to mostly likely increase the risk of preeclampsia in women with obesity and that these factors potentiate the antiangiogenic and proinflammatory mechanisms of placental ischemia-induced vascular dysfunction to increase the incidence of preeclampsia. [29] In the present study, the incidence of HDP or CH was significantly higher in women with obesity than in those without obesity among all women with T2DM (37.2% versus 16.1%, p = 0.0396). However, the incidence of HDP in women with obesity (BMI before pregnancy ≥ 25.0) was similar to that in women without obesity among the women with T1DM and among the women with T2DM without CH. In another study, multiple logistic regression analyses of pregnancy outcome among Japanese women with T1DM and T2DM revealed that the incidence of HDP (gestational hypertension or preeclampsia) and preeclampsia among women with T1DM (11.1% and 8.7%, respectively) were similar to those among women with T2DM (13.3% and 12.1%, respectively). The incidences of macrosomia, light-for-dates, and small for dates among infants of women

with T1DM (4.6%, 30.2%, and 11.7%, respectively) were similar to those among women with T2DM (5.0%, 32.8%, and 15.7%, respectively) [30]. In the present study, the mean birth weight among women with T1DM was significantly higher than that among those with T2DM; however, the incidence of FGR and the mean birth weight SD among women with T1DM were similar to that among women with T2DM.

Microvascular diseases including diabetic nephropathy are associated with high incidences of HDP, preeclampsia, and low birth weight [31]. In a previous study, the presence of diabetic vasculopathy and higher HbA1c and triglycerides levels in all three trimesters were associated with preeclampsia among women with T1DM [8]. In another study of 43 pregnant women with previous DM, 11 women with diabetic nephropathy exhibited higher incidences of CH (72.7% versus 21.9%, p = 0.004) and preeclampsia (63.6% versus 6.3%, p = 0.0003) and lower GW at delivery (36 weeks versus 38 weeks, p = 0.003) than 32 women without diabetic nephropathy [32]. In the present study, among the women with T1DM, the frequency of T1DM + HDP among women (n = 4, 66.7%) with diabetic nephropathy was significantly higher than that among their counterparts (n = 6, 15.4%; p = 0.0164). However, we could not assess the relationship between HDP and FGR among women with both T1DM and diabetic nephropathy because no woman with T1DM presented with both HDP and FGR. Among women with T2DM, the frequency of T2DM + HDP was similar between women with and without diabetic nephropathy (33.3% versus 11.3%); furthermore, the frequencies of women with T2DM + HDP or CH were similar (50.0% versus 28.9%). A recent review determined that it is also critical to screen for and manage retinopathy and nephropathy, and blood pressure goals must be considered carefully because lower treatment thresholds may be required for women with nephropathy [2].

CH occurs in up to 5% of pregnancies, and pregnancies complicated by CH are at risk for increased FGR and perinatal death [33]. In the previous cohort, T1DM, previous preeclampsia, and CH were strong risk factors for severe and preterm preeclampsia, and T1DM without other risk factors increased the risk of preterm preeclampsia, but not term preeclampsia [34]. In a retrospective study of women with CH, the fetal size curves determined via ultrasonography were not different from population standards, whereas there was an excess of both light-for-dates and heavy-for-dates infants. [35] In another study, lower plasma glucose levels at 1-h of a 50-g glucose challenge test at 24–28 GW was associated with a higher rate of light-for-dates among infants of the women with CH, and the rate of small for dates among infants of women with both CH and high plasma glucose levels was similar to that among infants of pregnant women without CH but with high plasma glucose levels. [36] Thus, among pregnant women with CH, DM could not offset FGR. The treatment of DM before pregnancy could undoubtedly prevent CH among women with DM. Preconception planning is very important to avoid unintended pregnancies and to minimize risk of congenital defects.

In the present study, 33.3% of women with T2DM+HDP had infants with FGR, and 55.6% of women with T2DM without CH who had infants with FGR had HDP. According to the previously mentioned two-stage model of preeclampsia, [12] these relationships were not surprising. However, these relationships were not found among women with T1DM. Among women with T1DM, HbA1c levels both during the first trimester and at delivery in women with HDP were significantly higher than those in women without HDP. Thus, DM with poor control might offset HDP-induced FGR.

The management of T1DM of pregnant women is important both before and during pregnancy. A recent review of women with T1DM indicated that preconception planning is extremely important for avoiding unintended pregnancies and minimizing the risk of fetal congenital abnormalities. The target HbA1c levels are <6.5% at conception and <6.0% during pregnancy. During pregnancy, continuous glucose monitoring can improve glycemic control

and neonatal outcomes in women with T1DM [2]. Another review of pregnant women with T1DM indicated that the goals of preconception care are tight glycemic control with an HbA1c level of <7.0% and as close to 6.0% as possible, and the target HbA1c level during pregnancy is <6.0%. In addition, the data regarding continuous glucose monitoring is conflicting regarding the improvement of glycemic control [37]. However, in previous reviews, despite significant reductions in serious adverse perinatal outcomes for women with T1DM during pregnancy, the opposite effect was observed on fetal overgrowth and associated complications. Pregnant women with T1DM who seemingly achieve adequate glycemic control during pregnancy often exhibit strongly fluctuating glycemic variability and sometimes continue to experience a greater risk of excess fetal growth, leading to heavy-for-dates infants and macrosomia [38]. Thus, several key points in the management of T1DM during pregnancy remain to be clarified. In Japan, the goals of DM management (T1DM and T2DM) before and during pregnancy focus on maintaining the targeted plasma glucose levels (<100 mg/dL before meals and <120 mg/dL after meals) opposed to maintaining the targeted HbA1c levels (<6.2% and as close to 6.0% as possible) [24]. In the present study, the frequencies of HbA1c levels <6.2% at delivery were higher than those before pregnancy among 45 women with T1DM (13.3% versus 27.5%, p = 0.0446) and 109 women with T2DM (33.3% versus 56.9%, p < 0.0001). However, the frequency of HbA1c levels <6.2% at delivery was higher among women with T2DM than among women with T1DM (p = 0.0126). The management of T1DM during pregnancy is a difficult future task.

In a retrospective cohort study of 375 singleton pregnant women with T1DM [39], the median HbA1c levels in the first, second, and third trimesters among 85 women (22.7%) with macrosomic infants were 6.4, 5.7, and 5.6%, respectively. There was a linear relationship between third-trimester HbA1c levels and macrosomia risk in the HbA1c range of 4.5%–7.0%. Macrosomia in infants born to women with T1DM was common despite excellent metabolic control. Conversely, in a retrospective chart review study of 308 singleton pregnancies in 221 women with T1DM and 87 women with T2DM, the rates of light-for-dates infants were 50% among women with T1DM and 23% among women with T2DM, and second-semester HbA1c levels among women with T1DM represented a risk factor for light-for-dates infants in a multivariable regression model [40].

In a retrospective observational study of 77 singleton pregnant women with T1DM in Japan [41], the insulin dosage gradually increased during pregnancy, reaching a maximum dosage at 35 GW that was 1.6fold higher than that prior to pregnancy. A significant negative correlation was observed between the insulin dosage increase and the duration of DM via multiple regression analyses. Women with a longer duration of DM required smaller increases of the insulin dosage during pregnancy, suggesting that a long duration of DM may decrease placental function. In another retrospective cohort study of 222 pregnant women with DM [42], women with T1DM (n = 67) required a significant higher insulin dosage in the first and second trimesters than those with T2DM (n = 155), but the insulin dosage in late pregnancy was similar between women with T1DM and T2DM. Women with T2DM required significantly greater increases in insulin dosage per trimester than those with T1DM. Women with T1DM, but not those with T2DM, had a significant decrease in the insulin dosage.

In a previous study, a composite neonatal outcome consisting of one or more early complications (respiratory distress, necrotizing enterocolitis, sepsis, transfusion, ventilation, seizure, hypoxic-ischemic encephalopathy, phototherapy, or death) was worse in women with hypertensive FGR than in those with normotensive FGR (50% versus 16%, p < 0.001), and a higher rate of maternal placental vascular lesions was detected in women with hypertensive FGR than in those with normotensive FGR (82% versus 58%, p < 0.001) [43].

The placenta plays a key role in sustaining fetal growth and development, and poorly controlled DM or pronounced obesity may exceed the placental homeostatic capacity, which has potentially adverse consequences for the fetus [44]. In a previous retrospective cohort of 302 pregnancies in women with T1DM, there was a significant positive association between placental weight and the risk for light-for-dates infants, which was particularly evident in pregnancies featuring poor glycemic control during the first trimester (especially first-trimester HbA1c levels ≥8.5%), highlighting the importance of achieving good glycemic control during early pregnancy [45]. Compared with the findings in women without preeclampsia, early-onset preeclampsia had significant associations with a lower weight, length, and width of the placenta independent of the duration of gestation and birth weight [46]. In pregnant women with DM and HDP, fetal growth might be determined by the balance between the increased blood glucose supply from the mother to their fetus via the heavier placenta according to the hyperglycemia–hyperinsulinemia theory [5] and the decreased blood glucose supply due to placental dysfunction in women with HDP according to the two-stage model for HDP [12].

Low-dose aspirin therapy has been used during pregnancy, most commonly to prevent or delay the onset of preeclampsia. Low-dose aspirin reduces thromboxane production. Low-dose aspirin therapy from 11 to 14 GW until 36 GW decreased the incidence of preterm preeclampsia among pregnant women with a high risk of recurrent preterm preeclampsia comparison with the findings for placebo [47]. The American College of Obstetricians and Gynecologists issued the Hypertension in Pregnancy Task Force Report recommending daily low-dose aspirin beginning in the late first trimester for women with a history of early-onset preeclampsia and preterm delivery at less than 34 GW and for women with more than one prior pregnancy complicated by preeclampsia [48]. Abnormal placentation resulting in poor placental perfusion (i.e., placental insufficiency) is the most common pathology associated with FGR [49]. Thus, low-dose aspirin prophylaxis for preventing recurrent FGR is similarly not currently recommended for women without other risk factors for preeclampsia because of insufficient evidence in women with an isolated history of FGR. However, in women at risk of preeclampsia, prophylaxis with low-dose aspirin (particularly when initiated less than 16 GW) may reduce the risk of FGR [48]. In a recent review article, low-dose aspirin was recommended soon after 12 GW to minimize the risk of preeclampsia in women with T1DM [2]. Some obstetricians indicated that pregnancies complicated by T1DM and T2DM increase the risk of these complications, which are largely attributed to placental dysfunction. The study of the role of low-dose aspirin therapy in preventing preeclampsia that included a subgroup of pregnant women with preexisting DM failed to demonstrate a benefit among women with preexisting DM, and the women were all recruited in the second trimester [50]. In a recent phase III double-blinded, placebo-controlled randomized multicenter clinical trial conducted in Ireland, researchers examined the effect of low-dose aspirin therapy from the first trimester until 36 GW on perinatal outcome in women with established pre-pregnancy DM, hypothesizing that aspirin therapy will reduce complications mediated by placental dysfunction and improve perinatal outcomes [51]. Low-dose aspirin therapy to prevent the onset of preeclampsia in pregnant women with DM might decrease the onset of FGR as a second-order effect.

This study also had two limitations. First, the number of patients was small. In particular, the number of women with HDP+FGR among those with T1DM was very small. Second, it was a retrospective cohort study without a control group. Further research with future prospective case–control studies or prospective randomized studies is therefore required.

In the resent review article, the authors described that advances have been made in their understanding of the pathophysiology and clinical management of preeclampsia, but several research questions remain. [52] Furthermore, the condition of DM in pregnancy is similar to preeclampsia.

## Conclusions

Poor control of DM is linked to HDP and FGR. The insufficient treatment of DM before pregnancy increases the risk of HDP. Mothers of infants with FGR might have a remarkably lighter placenta and a considerably higher insulin dosage at delivery than expected because of the placental dysfunction caused by HDP.

## Supporting information

**S1 Data. Women with DM data.**
(XLSX)

## Acknowledgments

The authors thank Enago (www.enago.jp) for the English language review.

## Author Contributions

**Conceptualization:** Hidemichi Watari.

**Data curation:** Mamoru Morikawa, Emi Kato-Hirayama, Michinori Mayama, Yoshihiro Saito, Kinuko Nakagawa, Takeshi Umazume, Kentaro Chiba, Satoshi Kawaguchi.

**Formal analysis:** Mamoru Morikawa.

**Supervision:** Kazuhiko Okuyama, Hidemichi Watari.

**Writing – original draft:** Mamoru Morikawa.

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
