## [Decision Letter · Decision Letter 0]

21 Jan 2020

PONE-D-19-34650

Glycemic control and fetal growth of women with both diabetes mellitus and hypertensive disorders of pregnancy

PLOS ONE

Dear Dr Morikawa,

Thank you for submitting your manuscript to PLOS ONE. After careful consideration, we feel that it has merit but does not fully meet PLOS ONE’s publication criteria as it currently stands. Therefore, we invite you to submit a revised version of the manuscript that addresses the points raised during the review process.

SPECIFIC ACADEMIC EDITOR COMMENTS. There were several major concerns raised by the reviewer that must be addressed. These relate to the need for extensive editing of English grammar and syntax; the abstract and introduction need to be significantly revised to provide more appropriate rationale/background information; the methods need to be strengthened; and the discussion needs to better present the current findings in relationship to more discussion of the published literature.

We would appreciate receiving your revised manuscript by Mar 06 2020 11:59PM. To enhance the reproducibility of your results, we recommend that if applicable you deposit your laboratory protocols in protocols.io, where a protocol can be assigned its own identifier (DOI) such that it can be cited independently in the future. For instructions see: http://journals.plos.org/plosone/s/submission-guidelines#loc-laboratory-protocols

We look forward to receiving your revised manuscript.

Kind regards,

Frank T. Spradley

Academic Editor

PLOS ONE

2. Please provide additional details regarding participant consent. In the ethics statement in the Methods and online submission information, please ensure that you have specified what type of consent you obtained (for instance, written or verbal). If your study included minors under age 18, state whether you obtained consent from parents or guardians. If the need for consent was waived by the ethics committee, please include this information.

"First authors received funding from Suzuki Diabetes Foundation (Tokyo, Japan) to publish this article (for the English language reviewing charge and article processing charge)."

Please remove any funding-related text from the manuscript and let us know how you would like to update your Funding Statement. Currently, your Funding Statement reads as follows: "No"

4. Thank you for stating the following in your Competing Interests section:  "No"

(Note: HTML markup is below. Please do not edit)

Reviewers' comments:

Reviewer's Responses to Questions

**Comments to the Author**

1. Is the manuscript technically sound, and do the data support the conclusions?

Reviewer #1: Yes

2. Has the statistical analysis been performed appropriately and rigorously? 

Reviewer #1: Yes

3. Have the authors made all data underlying the findings in their manuscript fully available?

Reviewer #1: Yes

4. Is the manuscript presented in an intelligible fashion and written in standard English?

Reviewer #1: No

5. Review Comments to the Author

Reviewer #1: Mamoru Morikawa et al tried to clarify the To clarify the relationship between glycemic control and fetal growth restriction (FGR) in women with diabetes mellitus (DM) and/or hypertensive disorder of pregnancy in this study. The following changes shall be made for further clarity.

1. Mandatory -Extensive English language edition is must and the paper should be prepared as per the journal instruction.

2. The title of the manuscript should be amended. What is the meaning of "both diabetes mellitus" and it is not a scientific word.

3. Abstract -Background of the study should be included and Methods section should be improved.

4. Introduction -This section is poorly presented.

5. Materials and Methods: Study design and inclusion and exclusion criteria of the study should be included (In a separate section

6. Discussion: More scientific evidences should be included

6. Conclusions should be rephrased.

6. PLOS authors have the option to publish the peer review history of their article (what does this mean?). If published, this will include your full peer review and any attached files.

Reviewer #1: No

---

## [Author Response · Author response to Decision Letter 0]

10 Feb 2020

We thank you for your constructive suggestions, which have helped us to improve our manuscript. We have revised the text accordingly and the modifications are indicated in blue. Our point-by-point responses to the reviewer comments have been provided.

PONE-D-19-34650

Glycemic control and fetal growth of women with both diabetes mellitus and hypertensive disorders of pregnancy

PLOS ONE

Dear Dr Morikawa,

Thank you for submitting your manuscript to PLOS ONE. After careful consideration, we feel that it has merit but does not fully meet PLOS ONE’s publication criteria as it currently stands. Therefore, we invite you to submit a revised version of the manuscript that addresses the points raised during the review process.

Response: Thank you for your consideration.

SPECIFIC ACADEMIC EDITOR COMMENTS. There were several major concerns raised by the reviewer that must be addressed. These relate to the need for extensive editing of English grammar and syntax; the abstract and introduction need to be significantly revised to provide more appropriate rationale/background information; the methods need to be strengthened; and the discussion needs to better present the current findings in relationship to more discussion of the published literature.

Response: We have revised the text per your remarks and modifications are indicated in blue. Our point-by-point responses to the reviewer comments have been provided.

We would appreciate receiving your revised manuscript by Mar 06 2020 11:59PM. To enhance the reproducibility of your results, we recommend that if applicable you deposit your laboratory protocols in protocols.io, where a protocol can be assigned its own identifier (DOI) such that it can be cited independently in the future. For instructions see: http://journals.plos.org/plosone/s/submission-guidelines#loc-laboratory-protocols

• A rebuttal letter that responds to each point raised by the academic editor and reviewer(s). This letter should be uploaded as separate file and labeled 'Response to Reviewers'.

• A marked-up copy of your manuscript that highlights changes made to the original version. This file should be uploaded as separate file and labeled 'Revised Manuscript with Track Changes'.

• An unmarked version of your revised paper without tracked changes. This file should be uploaded as separate file and labeled 'Manuscript'.

Response: We have submitted the “Response to Reviewers,” “Revised Manuscript with Track Changes,” and “Manuscript” files according to your comment.

Response: Tthank you for the comments.

We look forward to receiving your revised manuscript.

Response: We appreciate your comments.

Kind regards,

Frank T. Spradley

Academic Editor

PLOS ONE

1. Please ensure that your manuscript meets PLOS ONE's style requirements, including those for file naming. The PLOS ONE style templates can be found at http://www.plosone.org/attachments/PLOSOne_formatting_sample_main_body.pdf

 and http://www.plosone.org/attachments/PLOSOne_formatting_sample_title_authors_affiliations.pdf

Response: Accordingly, we have edited it according to the PLOS ONE style templates.

2. Please provide additional details regarding participant consent. In the ethics statement in the Methods and online submission information, please ensure that you have specified what type of consent you obtained (for instance, written or verbal). If your study included minors under age 18, state whether you obtained consent from parents or guardians. If the need for consent was waived by the ethics committee, please include this information.

Response: 

We have added ethical approval information in the Materials and methods as follows:

All women provided verbal informed consent to participate in the study. The information of this clinical study is released to the public via the website of Hokkaido University Hospital according to the recommendations of the Ministry of Health, Labor, and Welfare (Japan).

Fortunately, all 154 women were at least 20 years old at the time of delivery.

"First authors received funding from Suzuki Diabetes Foundation (Tokyo, Japan) to publish this article (for the English language reviewing charge and article processing charge)."

Please remove any funding-related text from the manuscript and let us know how you would like to update your Funding Statement. Currently, your Funding Statement reads as follows: "No"

Response: We apologize for our error. We have edited the text according to your suggestion. We have removed our “Funding Statement” from the Acknowledgments section and uploaded it to the online submission form.

4. Thank you for stating the following in your Competing Interests section: "No"

Response: We have added the following statement in our revised cover letter: “The authors have declared that no competing interests exist.” In addition, we have updated the online submission form accordingly.

Response: We could not obtain DOIs. We have uploaded our data set and also added the data set in the revised cover letter and revised manuscript.

Supporting information

S1 File. Women with DM data. Data set (XLSX).

Response: We have no ethical or legal restrictions to share our data publicly. We have added our Data Availability statement in the revised cover letter per your suggestion.

(Note: HTML markup is below. Please do not edit)

Reviewers' comments:

Reviewer's Responses to Questions

Comments to the Author

1. Is the manuscript technically sound, and do the data support the conclusions?

Reviewer #1: Yes

Response: Thank you for the comments.

2. Has the statistical analysis been performed appropriately and rigorously? 

Reviewer #1: Yes

Response: Thank you for the comments.

3. Have the authors made all data underlying the findings in their manuscript fully available?

Reviewer #1: Yes

Response: Thank you for the comments.

4. Is the manuscript presented in an intelligible fashion and written in standard English?

Reviewer #1: No

Response: Our revised manuscript has been submitted for English language review to Enago (www.enago.jp).

5. Review Comments to the Author

Reviewer #1: Mamoru Morikawa et al tried to clarify the To clarify the relationship between glycemic control and fetal growth restriction (FGR) in women with diabetes mellitus (DM) and/or hypertensive disorder of pregnancy in this study. The following changes shall be made for further clarity.

Response: Thank you for the comments. We have edited our manuscript according to your comments.

1. Mandatory -Extensive English language edition is must and the paper should be prepared as per the journal instruction.

Response: Our revised manuscript has been submitted for English language review to Enago (www.enago.jp).

2. The title of the manuscript should be amended. What is the meaning of "both diabetes mellitus" and it is not a scientific word.

Response: We have removed “both” and added “subsequent” to the title of the revised manuscript as follows: “Glycemic control and fetal growth of women with diabetes mellitus and subsequent hypertensive disorders of pregnancy”

3. Abstract -Background of the study should be included and Methods section should be improved.

Response: We have edited the abstract to <300 words and added the background of the study before the aim as follows: “We aim to clarify the relationship between glycemic control and fetal growth in women with DM and/or subsequent HDP.”

Background of the study: “Pregnant women with diabetes mellitus (DM) are at high risk for hypertensive disorder of pregnancy (HDP). Women with poor control DM sometimes have heavy-for-dates (HFD) infants. However, women with HDP sometimes have light-for-dates infants.”

We have edited the methods of the abstract as follows: 

“Of 7893 women with singleton births at or after 22 GW, we enrolled 154 women with type 1 (T1DM) or type 2 DM (T2DM) whose infants did not have fetal abnormalities. Among women with T1DM or T2DM, the characteristics of the three groups…”

4. Introduction -This section is poorly presented.

Response: We have added new details in Introduction section accordingly.

5. Materials and Methods: Study design and inclusion and exclusion criteria of the study should be included (In a separate section)

Response: We have included the “Study design” and “Inclusion and exclusion criteria” in the Materials and methods sections as follows:

Study design

 This retrospective cohort study was conducted in the perinatal medical centers of two institutions with maternal–fetal intensive care and neonatal intensive care units. The two institutions (Hokkaido University Hospital and Sapporo City Hospital) are located in Sapporo City with a population of 1.97 million and both institutions have specialists (medical doctors) in perinatal medicine and DM. These physicians worked in the Department of Obstetrics and Gynecology or the Department of Internal Medicine II, Hokkaido University Graduate School of Medicine.

Inclusion and exclusion criteria

Between January 2011 and December 2018, 7893 women gave birth at or after 22 GW at the two institutions. Of these, 154 women with DM who had singleton births were enrolled. Of these, those with “overt DM during pregnancy” [19-21] were included as having T2DM. Fortunately, all women were at least 20 years old at the time of delivery. Women whose infants had fetal chromosomal abnormality were excluded as well as those who gave birth to twins or triplets.

6. Discussion: More scientific evidences should be included

Response: We have provided new information in the Discussion section accordingly.

6. Conclusions should be rephrased.

Response: We have edited this section as follows:

Conclusions

Poor control of DM is linked to HDP and FGR. The insufficient treatment of DM before pregnancy increases the risk of HDP. Mothers of infants with FGR might have a remarkably lighter placenta and require a considerably higher insulin dosage at delivery than expected because of the placental dysfunction caused by HDP.

6. PLOS authors have the option to publish the peer review history of their article (what does this mean?). If published, this will include your full peer review and any attached files. Do you want your identity to be public for this peer review? For information about this choice, including consent withdrawal, please see our Privacy Policy.

Reviewer #1: No

Response: We agree with your response.

Response: Thank you for the comments. We have resubmitted our manuscript according to your comments.

While revising your submission, please upload your figure files to the Preflight Analysis and Conversion Engine (PACE) digital diagnostic tool,

https://pacev2.apexcovantage.com/. 

PACE helps ensure that figures meet PLOS requirements. To use PACE, you must first register as a user. Registration is free. Then, login and navigate to the UPLOAD tab, where you will find detailed instructions on how to use the tool. If you encounter any issues or have any questions when using PACE, please email us at figures@plos.org. Please note that Supporting Information files do not need this step.

Response: We have uploaded our figures to PACE.

---

## [Decision Letter · Decision Letter 1]

3 Mar 2020

Glycemic control and fetal growth of women with diabetes mellitus and hypertensive disorders of pregnancy

PONE-D-19-34650R1

Dear Dr. Morikawa,

We are pleased to inform you that your manuscript has been judged scientifically suitable for publication and will be formally accepted for publication once it complies with all outstanding technical requirements.

With kind regards,

Frank T. Spradley

Academic Editor

PLOS ONE

Reviewers' comments:

Reviewer's Responses to Questions

**Comments to the Author**

1. If the authors have adequately addressed your comments raised in a previous round of review and you feel that this manuscript is now acceptable for publication, you may indicate that here to bypass the “Comments to the Author” section, enter your conflict of interest statement in the “Confidential to Editor” section, and submit your "Accept" recommendation.

Reviewer #1: All comments have been addressed

2. Is the manuscript technically sound, and do the data support the conclusions?

Reviewer #1: Yes

3. Has the statistical analysis been performed appropriately and rigorously? 

Reviewer #1: Yes

4. Have the authors made all data underlying the findings in their manuscript fully available?

Reviewer #1: Yes

5. Is the manuscript presented in an intelligible fashion and written in standard English?

Reviewer #1: Yes

6. Review Comments to the Author

Reviewer #1: Author tried to address all the comments. There is a need of correcting typographical or grammatical errors.

7. PLOS authors have the option to publish the peer review history of their article (what does this mean?). If published, this will include your full peer review and any attached files.

Reviewer #1: No

---

## [Editor Report · Acceptance letter]

5 Mar 2020

PONE-D-19-34650R1 

Glycemic control and fetal growth of women with diabetes mellitus and hypertensive disorders of pregnancy 

Dear Dr. Morikawa:

I am pleased to inform you that your manuscript has been deemed suitable for publication in PLOS ONE. Congratulations! Your manuscript is now with our production department. 

With kind regards,

on behalf of

Dr. Frank T. Spradley 

Academic Editor

PLOS ONE